# Multiomics Approach Captures Hepatic Metabolic Network Altered by Chronic Ethanol Administration

**DOI:** 10.3390/biology12010028

**Published:** 2022-12-23

**Authors:** Isin Tuna Sakallioglu, Bridget Tripp, Jacy Kubik, Carol A. Casey, Paul Thomes, Robert Powers

**Affiliations:** 1Department of Chemistry, University of Nebraska-Lincoln, Lincoln, NE 68588-0304, USA; 2Center for Biotechnology, University of Nebraska-Lincoln, Lincoln, NE 68588-0665, USA; 3Nebraska Center for Integrated Biomolecular Communication, University of Nebraska-Lincoln, Lincoln, NE 68588-0304, USA; 4Department of Internal Medicine, University of Nebraska Medical Center, Omaha, NE 68198-5870, USA; 5Department of Biochemistry and Molecular Biology, University of Nebraska Medical Center, Omaha, NE 68198-5870, USA

**Keywords:** metabolomics, lipidomics, proteomics, alcoholic liver disease, hepatocytes, lipid droplets

## Abstract

**Simple Summary:**

The liver sustains the greatest damage from heavy alcohol consumption because alcohol is primarily detoxified in the liver. Accumulation of fat in the liver cells (hepatocytes) is one of the first pathological changes (fatty liver) that occurs in response to alcohol consumption. During continuous use/abuse of alcohol, fatty liver progresses to more severe forms of a liver disease such as fibrosis, cirrhosis, and cancer. In this study, we used several high-throughput biochemical assays that were combined during analysis (Multiomics) to determine molecular changes induced by alcohol during the development of fatty liver in an alcohol fed animal model. We found that alcohol administration increased several fatty acid species that are precursors to triglycerides (fat). Notably, while there was an increase in glucuronidation (biochemical modification of compounds for removal from body) of toxic metabolites, glucuronidation of cholesterols were selectively decreased after alcohol administration. These findings suggest that alcohol administration promotes precursors essential for fat formation and simultaneously decreases the removal of cholesterol to disrupt hepatic metabolic homeostasis. Our findings provide deeper insights into metabolic pathways altered by alcohol and forms the basis for further investigations that can reveal potential druggable targets to treat alcohol associated fatty liver disease.

**Abstract:**

Using a multiplatform and multiomics approach, we identified metabolites, lipids, proteins, and metabolic pathways that were altered in the liver after chronic ethanol administration. A functional enrichment analysis of the multiomics dataset revealed that rats treated with ethanol experienced an increase in hepatic fatty acyl content, which is consistent with an initial development of steatosis. The nuclear magnetic resonance spectroscopy (NMR) and liquid chromatography–mass spectrometry (LC-MS) metabolomics data revealed that the chronic ethanol exposure selectively modified toxic substances such as an increase in glucuronidation tyramine and benzoyl; and a depletion in cholesterol-conjugated glucuronides. Similarly, the lipidomics results revealed that ethanol decreased diacylglycerol, and increased triacylglycerol, sterol, and cholesterol biosynthesis. An integrated metabolomics and lipidomics pathway analysis showed that the accumulation of hepatic lipids occurred by ethanol modulation of the upstream lipid regulatory pathways, specifically glycolysis and glucuronides pathways. A proteomics analysis of lipid droplets isolated from control EtOH-fed rats and a subsequent functional enrichment analysis revealed that the proteomics data corroborated the metabolomic and lipidomic findings that chronic ethanol administration altered the glucuronidation pathway.

## 1. Introduction

Alcoholic liver disease (ALD) is a leading cause of mortality worldwide [1,2,3]. The world health organization (WHO) reports 3 million deaths every year from ALD, which represents 5.3% of all deaths [1]. ALD is actually a spectrum of disease states that ranges from asymptomatic stages to alcoholic fatty liver, and finally to complete liver failure [4]. An early manifestation of ALD is the development of hepatic steatosis or a fatty liver that can be reversed by alcohol abstinence [5,6]. However, a continuous consumption of alcohol can progress to alcoholic hepatitis, cirrhosis, and eventually death [4,7]. Thus, there is a critical need to understand the molecular mechanism of ALD in order to diagnose the disease at its earliest stages (i.e., steatosis) to prevent progression to the later stages and a fatal outcome. 

Chronic ethanol consumption increases the production of reactive oxygen species (ROS), lowers cellular antioxidant levels, and enhances oxidative stress in the liver [3,5]. During ethanol exposure, fats and lipids accumulate in the liver and react with free radicals and other secondary metabolites [8]. This triggers a sequence of reactions that damages biomolecules and organelles, which eventually disrupts the homeostasis of the hepatocytes during hepatic steatosis [8]. Hepatocytes are the major cell type within the liver and comprise 80% of its total mass [9,10]. Ethanol-induced fatty liver is a consequence of the dysregulation of several cellular mechanisms [11]. Hepatic ethanol metabolism lowers the hepatic NAD^+^/NADH ratio, which initiates significant metabolic shifts toward reductive synthesis, which accelerates the synthesis and slows the oxidation of fatty acids [12]. Therefore, this ethanol-induced homeostatic disruption causes hepatocytes to accumulate fats in the form of lipid droplets (LDs). LD accumulation occurs partly due to the activation of de novo lipogenesis [13,14]. LDs play an important role in the regulation of intracellular lipid storage and lipid metabolism of the hepatocytes during the early stages of ALD [15,16].

In the last few years, high throughput multiomics technologies have revolutionized biomedical research [17]. Obtaining multiple molecular measurements such as metabolomics, transcriptomics, and proteomics data sets from a single sample provides a unique system-wide view that is essential to better understand the disease state and to improve treatments. Simply, combining multiple omics techniques may uncover new biological insights that are not likely to be accomplished with a single approach [18]. Despite the on-going advancements in multiomics techniques, lipidomics, metabolomics and proteomics have never been applied together in a study to investigate ALD. In fact, only 8 studies over the last ten years have individually used either lipidomics, metabolomics, or proteomics to investigate liver injuries or ALD (Appendix A). For example, Beyoglu et al. (2020) identified some potential biomarkers of ALD by searching the scientific literature for clinical metabolomics studies involving patients with ALD, cholestasis, fibrosis, cirrhosis, nonalcoholic fatty liver, and nonalcoholic steatohepatitis [19]. A comprehensive analysis of this extensive collection of metabolic data suggested lipids such as triglycerides and phospholipids were altered in ALD patients and may serve as disease biomarkers. Itturospe et al. (2022) found a metabolic signature of alcohol induced hepatotoxicity that included ceramides, carnitines, acetylcholine, ethoxylated phosphocholine in HepaRG cells [20]. Fang et al. (2019) showed that an LC-MS metabolomics analysis of urine samples identified tricarboxylic acid (TCA) cycle metabolites and pentose phosphate glucoronate interconversion as being altered in an ALD rat model [21]. A similar metabolomics analysis of an ALD mouse model identified ethyl-beta-D-glucuronide, ethyl sulfate, and indole lactic acid as potential biomarkers [22].

Herein, we report the first multiomics investigation of EtOH-induced hepatocellular changes that combines a metabolomics, lipidomics, and proteomics approach. Our study highlights the beneficial impact of a combined multiplatform, multiomics approach in achieving a system-wide characterization of EtOH-induced hepatocellular changes. Specifically, we measured global metabolic and lipidomic changes in hepatocytes isolated from control and EtOH-fed rats to uncover major regulating factors of steatosis. We also investigated global membrane proteome changes in LDs isolated from chronic EtOH-fed rats and correlated the overall changes in the regulome of small molecules with these LD membrane proteome changes. While the global regulome from the integrated data set of small molecules showed fatty acyl as the most impacted subclass of metabolites, only the combined application of proteomics and metabolomics revealed glucuronidation was a highly perturbed metabolic pathway. We observed an increase in the glucuronidation of toxic substances, tyramine, and benzoyl, to increase their solubility for subsequent excretion. Conversely, we observed a decrease in cholesterol-conjugated glucuronides with a simultaneous increase in free cholesterol. This free cholesterol may induce toxicity, but as previously observed is likely safely sequestered into the LDs [23]. Both the accumulation of cholesterol in the LDs and the glucuronidation of toxic metabolites suggest a protective mechanism in response to ethanol exposure [24].

## 2. Materials and Methods

Primary rat hepatocytes from control (*n* = 7) and EtOH-fed animals (*n* = 6) were prepared from male Wistar rats (Charles River Laboratories, Wilmington, MA, USA). One EtOH-fed rat died during the alcohol feeding study. A power analysis identified 4-6 animals per group as the minimal sample size to provide a significance level at an α of 0.05. Lipid droplets were extracted from the rat hepatocytes prior to protein extraction. All animals were initially fed a basic Purina chow diet while acclimating to the surroundings during the first 3 days. The animals were then paired by weight and split into control-fed or EtOH-fed groups and maintained for 5–8 weeks. The EtOH-containing Lieber-DeCarli diet consisted of 35% fat, 18% protein, 11% carbohydrates, and 36% EtOH (Dyets Inc., Bethlehem, PA, USA). The control diet was identical to the EtOH diet except for the isocaloric substitution of carbohydrates in the place of the EtOH. This protocol was approved by the Institutional Animal Care and Use Committee of the Department of Veterans Affairs, Nebraska Western Iowa Health Care System, and the University of Nebraska Medical Center.

Lipids and metabolites were extracted from the hepatocytes using either an organic or aqueous solvent extraction protocol, respectively and then characterized by NMR and LC-MS. A UPLC system coupled to a Xevo G2-XS Q-TOF (Waters MS Technologies, Manchester, UK) equipped with an electrospray ionization (ESI) source operating in positive ionization mode was used for the LC-MS metabolomics and lipidomics experiments. A Bruker Avance III-HD 700 MHz spectrometer (Bruker BioSpin, Billerica, MA, USA) equipped with a quadruple resonance QCI-P cryoprobe with z-axis gradients, SampleJet sample changer with IconNMR and an autotune and match (ATM) were used for the NMR metabolomics study.

Data processing, multivariate and univariate statistical analysis, and metabolite and lipid identification of the metabolomics and lipidomics data sets relied on Chenomx NMR Suite 8.3 (Chenomx, Edmonton, AB, Canada), MetaboAnalyst 5.0 (https://www.metaboanalyst.ca/, accessed on 21 December 2022) [25], MVAPACK (https://mvapack.unl.edu/, accessed on 21 December 2022) [26], Progenesis^®^ QI metabolomics software (version 2.4, Nonlinear Dynamics, Newcastle, UK), and the Human Metabolome Database (HMDB 5.0, https://hmdb.ca/, accessed on 21 December 2022) [27]. Metabolites and lipids were identified using a combination of exact mass and MS^E^ fragmentation patterns corresponding to a putative or MSI level 2 assignment.

Proteomics analysis of the lipid droplets was accomplished using a Thermo Orbitrap Fusion Lumos Tribrid (Thermo Scientific™, Waltham, MA, USA) mass spectrometer in a data dependent acquisition mode. LC-MS data was analyzed with Progenesis QI proteomics 4.2 (Nonlinear Dynamics, Milford, MA, USA), and MS/MS data were annotated with the Swiss-Prot rat protein database using the in-house PEAKS X + DB search engine. Cytoscape v 3.9.0 with ClueGo v 2.5.8 plugin [28,29] combined with a manual analysis was used for a network analysis of the combined or individual omics datasets. A detailed description of the methods and materials are provided in the Appendix A [25,26,28,29,30,31,32,33,34,35,36,37,38,39,40,41].

## 3. Results

### 3.1. Ethanol Exposure Induced LD Accumulation in Hepatocytes

Hematoxylin and eosin (H&E) staining of liver sections showed a significant accumulation of lipids in the livers of ethanol-fed rats, as judged by the presence of unstained, sharply defined cytoplasmic vacuoles (Figure 1A,B). Oil red O staining of hepatocytes isolated from the experimental animals revealed that compared to controls, hepatocytes from ethanol-fed rats exhibited a 2- and 3-fold increase in the number and size of LDs, respectively (Figure 1 C,D).

### 3.2. Ethanol Exposure Induces a Distinct Alteration in the Metabolome and Lipidome of Hepatocytes

The hepatocytes from the liver of ethanol exposed rats were fractionated and extracted for lipidomics and metabolomics analysis. Specifically, both one-dimensional (1D) ^1^H NMR and LC-MS analytical techniques were used to measure metabolic changes but only LC-MS was used to measure lipid changes. A principal component analysis (PCA) model was generated from the individual and combined omics datasets for a total of four PCA models. The individual PCA models were created with 289 NMR spectral features (Figure 2A),1136 LC-MS spectral features from the metabolomics dataset (Figure 2B), or 504 LC-MS spectral features from the lipidomics dataset (Figure 2C). The integrated PCA model (Figure 2D) was created from a total of 1929 spectral features by combining the NMR metabolomics dataset and the LC-MS metabolomics and lipidomics datasets. Specifically, the metabolomics and lipidomics datasets were integrated into a single data matrix and analyzed with the bio-conductor package in R [42], MetaboAnalyst 5.0 [25], and our MVAPACK [26] chemometric toolkit. The scaled, transformed, normalized data matrix of 1929 features showed a valid normalized bell-shaped curve (Appendix A) indicating a normal distribution for the integrated matrix.

All four PCA scores plots exhibited similar and significant differences between the control and EtOH-fed groups with high statistical quality metrics as indicated by average *R^2^* and *Q^2^* > 0.7 ± 0.2. The PCA model using the NMR dataset had the lowest quality metrics while the LC-MS lipidomics dataset had the best metrics. Overall, the *Q^2^* metrics indicate that the PCA is a consistent model with a good fit to the data and a reliable indicator of a group separation. The *R^2^* and *Q^2^* values for each of the four individual PCA models are listed in Appendix A.

### 3.3. The Integrated Omics Dataset Identified Fatty Acyls and Glycerophospholipids as Being Altered by Ethanol

A supervised orthogonal projection to latent structures-discriminant analysis (OPLS-DA) model was generated from the combined metabolomics and lipidomics dataset. The resulting OPLS-DA scores plot showed a clear separation between the control and EtOH-fed groups (Appendix A). The OPLS-DA model yielded high statistical quality metrics given an *R^2^* of 0.996 and a *Q^2^* of 0.994 and was validated using a permutation test (*n* = 1000) that yielded a *p*-value of 0.001 (Appendix A). The OPLS-DA model also identified the key spectral features (i.e., variable importance in projection (VIP) features and scores) that defined the separation between the control and EtOH-fed groups. The significantly altered VIP features were defined as having a false-discovery rate (FDR) corrected *p*-value < 0.01, a fold change > 1.5, and a VIP > 1.19. The top 11 metabolites identified from these significantly altered VIP features are listed in Table 1. These metabolites were assigned to the following chemical classes: fatty acyls, glycerophospholipids, organooxygen compounds, pteridines and derivatives, and saccharolipids. Notably, five (45%) of the top 11 metabolites are fatty acyls. The metabolite subclasses are also listed in Table 1.

Representative box plots are shown in Figure 3, which clearly indicates the statistically significant increase or decrease in metabolite concentrations that resulted from EtOH feeding. For example, EtOH increased hydroxy isovaleric acid, methyl-butenoyl-apiosylglucose, and gingerdiol-beta-glucopyranoside, which are in the fatty acyl glycosides or fatty acids and conjugates subclasses of the fatty acyls class (Figure 3A). The two remaining fatty acyls, tricosenoic acid and dimethylnonanoyl carnitine, decreased following EtOH feeding and belong to two different subclasses, unsaturated fatty acids, and fatty acid esters, respectively. Two other metabolites, PI (36:4) and LysoPA (18:3), belong to the glycerophospholipids class. EtOH feeding decreased PI (36:4), but increased LysoPA (18:3) (Figure 3B). Glucose-1-phosphate belongs to the organooxygen compound class and was downregulated after alcohol feeding. The last two VIP metabolites, riboflavin, and UDP-O-beta hydroxymyristoyl-GlcNac, were assigned to the pteridines and derivatives, and saccharolipids classes, respectively. EtOH feeding increased riboflavin but decreased UDP-O-beta hydroxymyristoyl-GlcNac (Figure 3C,D).

### 3.4. The Metabolomics Datasets Identified Glucuronides as Being Altered by Ethanol

Two individual OPLS-DA models were created from either the NMR or LC-MS metabolomics datasets. The lipidomics data set was omitted from this analysis. The two pair-wised OPLS-DA models showed a distinct and statistically significant separation between the control and EtOH-fed groups. The OPLS-DA model yielded high statistical quality metrics given an average *R^2^* and *Q^2^* of 0.98 ± 0.01 and 0.97 ± 0.03, respectively (Appendix A). The OPLS-DA models were validated with a permutation test (*n* = 1000) that resulted in an average *p*-value of 0.001. A univariate FDR corrected *p*-value < 0.001 and a fold change >1.5 were used to identify statistically significant metabolite changes. A total of 76 metabolites were identified that differed between the control and EtOH-fed groups from either the LC-MS or NMR datasets (Appendix A). An enrichment analysis of these 76 metabolites with MetaboAnalyst identified several metabolic subclasses that were affected by EtOH administration (Figure 4A). The top five EtOH-altered metabolic pathways based on an FDR-corrected *p*-value < 0.001 were glucuronides, amino acids, leukotrienes, and monosaccharides (Figure 4A, Appendix A). Notably, the exclusion of the lipidomics dataset from this analysis caused the fatty acyls and phospholipids to no longer be among the top altered subclasses.

A few select representative box plots for metabolites altered by EtOH administration are shown in Figure 4B–E. Again, it is clear from these box plots that EtOH feeding induced a statistically significant increase or decrease in the concentration of a specific set of metabolites. For example, EtOH administration decreased sterol-conjugated glucuronides such as hydroxy androsterone-glucuronide and dehydrotestosterone-glucuronide and increased other glucuronides such as tyramine-glucuronide and benzoyl glucuronide (Figure 4B). EtOH feeding also led to an increase in several amino acids like alanine and lysine (Figure 4C). For leukotrienes, EtOH induced an increase in oxo-dihydroxy-leukotriene B4, but it led to a decrease in leukotriene E4 (Figure 4D). Among the monosaccharides, xylose and xylulose were both increased upon EtOH administration (Figure 4E). While fatty acyl glycosides and fatty acyl carnitines (i.e., subclasses of fatty acyl) were still identified by the enrichment analysis, they were not a top choice in the absence of the lipidomics dataset due to the limited number of metabolites observed as a hit (Appendix A).

### 3.5. Lipidomics Reveals Lipids, Sterols, and Cholesterols Species Are Dysregulated by EtOH

A supervised OPLS-DA model was generated from only the lipidomics dataset. As before, the NMR and LC-MS metabolomics datasets were excluded from this analysis. The pair-wised OPLS-DA model showed a distinct and statistically significant separation between the control and EtOH-fed groups (Appendix A). The OPLS-DA model yielded high statistical quality metrics given an *R^2^* of 0.997 and *Q^2^* of 0.996. The OPLS-DA model was validated with a permutation test (*n* = 1000) that resulted in a *p*-value of 0.001 (Appendix A). A univariate FDR corrected *p*-value < 0.001 and a fold change >2 were used to identify statistically significant metabolite changes. A total of 45 lipids from the lipidomics dataset were identified that differed between the control and EtOH-fed groups (Appendix A). An enrichment analysis with MetaboAnalyst identified several metabolic subclasses based on 39 out of the 45 lipids that were affected by EtOH administration (Figure 5A). The top enriched lipid classes (*p* < 0.001) included diacylglycerol, triglycerols, thia fatty acids, diacylaminosugars, diacylglycosylglycerophospholipids and eicosanoids. Specifically, EtOH feeding increased the concentration of DG (36:3) from the diacylglycerol class (Figure 5B), mercaptooctadecanoic acid from thia fatty acids (Figure 5C), TG (60:1) and TG (50:0) from the triacylglycerol class (Figure 5D), 20:3-Glc-stigmasterol from the stigmasterol class, and beta-hydroxy-alpha-methyl-alpha-cholestene-carboxylic acid from the cholesterol class (Figure 5E,F). It is noteworthy that stigmasterol is a plant sterol and they are implicated in cholesterol and bile acid excretion, and tissue injury [43,44,45]. The Lieber DeCarli diet used in this study was composed of fat, carbohydrate, and protein, where the sole source of fat was Olive oil. Since Olive oil is a good source of plant sterol, the consumption of Lieber DeCarli diet was expected to result in the hepatic occurrence of stigmasterol. Alcohol consumption was postulated to either slow rat metabolism or to reduce excretion, leading to an accumulation of stigmasterol in the hepatocytes. However, additional work is needed to determine the pathways affected by stigmasterol accumulation in the liver, and whether stigmasterol-dependent metabolic alteration is hepatoprotective or hepatotoxic.

### 3.6. Proteomics Identified Glucuronidation as Being Significantly Altered in LDs Following Ethanol Treatment

LDs isolated from whole livers were used to characterize changes in the LD membrane proteins induced by ethanol exposure. LC-MS was used to identify and quantify the proteins extracted from LDs isolated from the control and EtOH-fed rats. A total of 2083 proteins were identified using a label-free untargeted proteomics profiling technique. A volcano plot of these proteins is displayed in Figure 6A. A total of 368 proteins were differentially expressed due to EtOH feeding (FC > 2.0, *p*-value < 0.05), where 300 proteins were decreased, and 68 proteins were increased (Figure 6A). The complete list of differentially expressed proteins were submitted to Cytoscape (https://cytoscape.org/, accessed on 21 December 2022) to identify the enriched biological processes (Figure 6B). Of the 368 differentially expressed proteins, only 124 were enriched in 17 KEGG Pathways (Appendix A), in which 15 of these pathways were significantly enriched with a *p*-value < 0.05. These 15 significantly enriched pathways included oxidative phosphorylation (17 proteins), metabolic pathways (48 proteins), steroid hormone biosynthesis (8 proteins), steroid biosynthesis (4 proteins), carbon metabolism (7 proteins), drug metabolism (5 proteins), ascorbate and aldarate metabolism (3 proteins), biosynthesis of amino acids (5 proteins), taurine and hypotaurine metabolism (2 proteins), starch and sucrose metabolism (3 proteins), pentose and glucuronate interconversion (3 proteins), fructose and mannose metabolism (3 proteins), tyrosine metabolism (3 proteins), linoleic acid metabolism (3 proteins), arachidonic acid metabolism (4 proteins). Notably, 7 of these top enriched pathways were also identified by the metabolomics and lipidomics data and are marked with a star in Figure 6B. The common pathways identified from the metabolomics and proteomics datasets are marked with red stars and correspond to biosynthesis of amino acids, pentose and glucuronate interconversion (i.e., glucuronidation), fructose and mannose metabolism. The common pathways identified from the lipidomics, and proteomics datasets are marked with green stars and correspond to steroid hormone biosynthesis, steroid biosynthesis, linoleic acid metabolism, and arachidonic acid metabolism. Oxidative phosphorylation, the top enriched pathway from the proteomics data set, was only identified in the proteome from the LDs, it was not observed in the metabolomics or lipidomics datasets. These 15 significantly enriched pathways and the proteins related to these pathways were submitted to Cytoscape to generate a protein–protein interaction (PPI) network (Figure 6C). The metabolomics, lipidomics, and proteomics data sets were not integrated for the Cytoscape analysis because the data sets were derived from two distinct biological samples, lipid drops and hepatocytes. The PPI network suggests UDP-glucuronosyltransferases (UGTs) and cytochrome proteins (Cyps) are highly networked and interact with a majority of the top enriched pathways.

We also analyzed these 368 differentially expressed proteins with another database, WikiPathways (https://www.wikipathways.org, accessed on 21 December 2022) [46]. In this case, only 47 were enriched in 6 WikiPathways (Appendix A), in which only 3 were statistically significant (*p*-value < 0.05) and corresponded to electron transport chain (16 proteins), oxidative phosphorylation (12 proteins) and glucuronidation pathways (4 proteins) (Appendix A). Both the KEGG and WikiPathway proteomics analysis and the metabolomics pathways analysis corroborate the significance of UGTs and glucuronides in response to EtOH exposure. The relative concentration changes in the UGT proteins are shown as box blots in Figure 6D. EtOH feeding upregulated the expression of 3 glucuronidation related proteins (Ugt-2b1, Ugt-2b10, Ugt-2b) and decreased the expression of one protein (Upg2) that are all essential to glucuronidation pathway (Figure 6D). Notably, a total of 12 UGT proteins were detected in the proteomics dataset corresponding to 6 Ugt1s (Ugt-1A1, Ugt-1A5, Ugt-1, Ugt-1A6, Ugt-1A9, Ugt-1A3) and 6 UGT2s. All the Ugt1S proteins and the remaining three UGT2s (Ugt-2b37, 2b17, 2a3) were not significantly altered with an average FC of 1.2 and a *p*-value > 0.1.

### 3.7. The mRNA Levels of Ugt-1a1 Were Decreased but No Change in the Enzyme Activity for Pan UGTs Was Observed Following Ethanol Exposure

Based on the outcome of the proteomics experiment, mRNA levels of *UGTs* were measured using a quantitative polymerase chain reaction (qPCR). The enzymatic activity of UGTs were measured using a commercial assay kit (ligand screening kit), which monitors the change in fluorescence of a UGT substrate. The mRNA expression level of *Ugt-1a1* was decreased in hepatocytes from ethanol exposed rat (Appendix A). Conversely, there was no observable change in the enzymatic activity of UGTs following ethanol exposure (Appendix A).

### 3.8. An Overall Network Map Shows Ethanol Administration Primarily Affected Glucuronides and Cholesterol Metabolism

The individual outcomes from the NMR metabolomics, LC-MS metabolomics, and LC-MS lipidomics datasets were combined to create a comprehensive metabolic and lipidomic network (Figure 7A). Specifically, discriminatory spectral features from each omics dataset were selected based on a univariate statistical analysis requiring a FC > 2.0 and an FDR-corrected *p* < 0.05. These selected features were then annotated to assemble a list of statistically significant metabolites and lipids for the network analysis. 33 metabolites were identified from the NMR dataset; and 43 metabolites and 45 lipids were identified from the LC-MS datasets. A total of 76 metabolites and 45 lipids were then submitted for enrichment pathway analysis using MetaboAnalyst. Of the 76 metabolites, only 42 metabolites were enriched in a specific metabolite subclass with an FDR-corrected *p*-value < 0.01 (Appendix A). A total of 11 metabolite subclasses were identified, where glucuronides was the most enriched subclass and comprised 26% of the metabolite enrichments. Of the 45 lipids, only 39 were enriched in a specific lipid subclass with an FDR-corrected *p*-value < 0.05 (Appendix A). A total of 16 lipid subclasses were identified, where diacylglycerols (DAGs) and triacylglycerol (TAGs) were the most enriched subclasses. DAGs and TAGs comprised 15% and 30% of the lipid enrichments, respectively.

A metabolomics and lipid network map (Figure 7A) was created by manually curating the statistically significant metabolites (MS, red; NMR, blue) and lipids (green) with an observed enrichment analysis (FDR-corrected *p* < 0.05) and adding them to a standard set of metabolic pathways that was then plotted with the Biorender application [47]. Metabolites and lipids that were increased due to ethanol treatment are shown as up arrows. Metabolites and lipids that were decreased due to ethanol treatment are shown as a down arrow. The network map was constructed sequentially by first starting with glucuronides, the most enriched subclass of metabolites, which were then followed by DAGs and TAGs, the most enriched subclass of lipids. The metabolic pathways upstream or downstream of these enriched metabolites and lipids were then added sequentially to the network map. For example, the TCA cycle is upstream from both triglycerides and glucuronides and TCA cycle intermediates, citrate, and succinate (FC > 3.0, FDR-corrected *p* < 0.001, VIP > 1.0), were also identified in the enrichment analysis. Thus, the TCA cycle along with glycolysis and gluconeogenesis were added to the network map. Similarly, pyruvate, which is the end-product of the glycolysis pathway and feeds into the TCA cycle and fatty acids pathways, was also added to the network map. Pyruvate was significantly increased in ethanol treated hepatocytes (FC > 2.79, FDR-corrected *p* < 0.001, VIP = 0.85). Other metabolites and lipids were added to the network map in a similar manner.

We also created a summary network map that combines the metabolomics, lipidomics, proteomics data (Figure 7B). The summary network map was created by manually curating the statistically significant metabolites (MS, red; NMR, blue), lipids (green), and LD proteins (purple) with an observed enrichment analysis (FDR-corrected *p* < 0.05). Again, this summary network map was sequentially constructed by first starting with the glucuronidation pathway, which comprised the most enriched subclass of metabolites. The UGT proteins, which are responsible for glucuronidation interconversion and were one of the most enriched pathways from the proteomics dataset, were then added to the network. The cholesteryl ester, which is one of the most enriched subclasses of lipids, and the proteins related to steroid biosynthesis, a significant class from the proteomics dataset that also interconnects with the cholesteryl ester biosynthesis, were then added to network. Finally, the metabolic pathways upstream or downstream of these enriched subclasses of metabolites and lipids were added to the network map. For instance, the fatty acids and TCA cycle pathways, which are upstream of cholesterol and glucuronides and contain enriched metabolites and lipids were placed on the summary network map. Other metabolites, lipids, and proteins were added to the summary network map in a similar manner.

## 4. Discussion

### 4.1. Ethanol Exposure Induced LD Accumulation and Metabolic Changes

Fatty liver (steatosis), which is characterized by the accumulation of LDs in hepatocytes, is one of the earliest cellular events that takes place after alcohol consumption [11,20]. The first and most impactful phenotypical change of hepatocytes during ethanol exposure is LD accumulation. Here, we observed the accumulation of LDs in the livers of EtOH-fed rats. Specifically, stained hepatocyte cells were observed to have a high number of LDs (Figure 1). A multiplatform and multiomics approach was then employed to provide a comprehensive characterization of the key molecular changes that occur in hepatocytes exposed to EtOH to better understand the development of EtOH-induced hepatocellular changes and fatty liver. To the best of our knowledge, this is the first time a multiplatform and multiomics approach was used to systematically investigate the molecular mechanism of EtOH-induced hepatocellular changes and fatty liver.

Hepatocytes from the liver of ethanol exposed rats and control animals were lysed and solvent extracted for lipidomics and metabolomics analysis using both NMR and LC-MS. Similarly, the LDs isolated from whole livers of control and EtOH-fed rats were used for a proteomics analysis. We individually analyzed each omics data sets, but more importantly, we also integrated the LC-MS lipidomics, LC-MS metabolomics, and NMR metabolomics data sets into a single combined matrix to unravel the metabolite and lipid classes in the hepatocytes cells that were most impact by EtOH [25,40]. Furthermore, we manually curated the metabolomics, lipidomics and proteomics data to identify the most impacted and interconnected pathways to generate a consensus system-wide view of the cellular response of hepatocytes to EtOH treatment.

### 4.2. Ethanol Exposure Induced Changes to the Fatty Acid Content of Hepatocytes

PCA and OPLS-DA models of the individual or combined lipidomics and metabolomics datasets showed a statistically significant difference between controls and EtOH exposed hepatocytes (Figure 2, Appendix A). A significant separation in the PCA scores plots implied an overall difference in the metabolomes and lipidomes between the two groups and warrants further investigation into the identification of the specific metabolites, lipids, and metabolic processes that are dysregulated due to EtOH. The OPLS-DA model of the integrated metabolomics and lipidomics data set identified fatty acyls as the lipid class most impacted by EtOH (Figure 3A, Table 1). This is consistent with previous reports that lipids and fatty acyls accumulate during the development of fatty liver disease. Other altered lipids were glycerophospholipids, organooxygen compounds, and saccharolipids (Figure 3B–D). Overall, the integrated omics data set indicted that 72% of the highly altered molecular subclasses were lipids

### 4.3. Ethanol Exposure Induced Hepatocytes to Sequester TAGs and Cholesterol in LDs

Given the significant changes in the fatty acyl family, we then analyzed the lipidomics data set separately. Accordingly, TAGs, DAGs, and thia fatty acids were identified as the top altered lipid subclasses (Figure 5A). We observed an increase in DAGs and thia fatty acids (Figure 5B,C), and a correlated increase in TAGs and cholesterol derivatives (Figure 5D–F). These lipid changes reflect the phenotypical accumulation of LDs, which are primarily composed of TAGs and cholesteryl esters [15,16].

### 4.4. Ethanol Induced the Hydroxylation of Fatty Acids

Fatty acyls subclasses, such as prostaglandins (PGF alpha methyl ether), thia fatty acids (mercapto-octadecanoic acid), and eicosanoids (hydroxy-eicosadienoic acid, HEDE), were likely accumulated in EtOH exposed hepatocytes due to oxidative stress [48,49]. Mercapto-octadecanoic acid was increased 10-fold in the ethanol exposed hepatocytes (Figure 5C). Thia fatty acids are known to increase fatty acid oxidation in the liver through the inhibition of malonyl-CoA synthesis, activation of carnitine palmitoyl transferase I (Cpt-I), and induction of Cpt-II and enzymes of peroxisomal beta-oxidation. We observed an increase in Cpt-I in ethanol exposed hepatocytes, which was consistent with the increase in thia fatty acids and an alteration in fatty acid oxidation (Appendix A). Activation of fatty acid oxidation is critical to the hypolipidemic effect of thia fatty acid [49]. The hydroxylation of fatty acids is commonly observed in steatosis due to the oxidative stress of ethanol exposure [50,51]. Accordingly, we observed an increase in the hydroxylated fatty acid HEDE in ethanol exposed hepatocytes.

### 4.5. Proteomics and Lipidomics Results Both Agree with the Accumulation of Cholesterol Synthesis in LDs

Recent evidence suggests that LDs enclosed by a phospholipid monolayer harbors LD proteins that control triglyceride and cholesterol metabolism. Therefore, we sought to determine if changes in hepatic lipid metabolism induced by EtOH feeding were correlated with LD membrane proteome changes. Consistent with the lipidomics results, our proteomics analysis of the LD membrane proteins identified proteins associated with fatty acid biosynthesis, including linoleic acid and arachidonic acid metabolism. Proteins associated with steroid and steroid hormone biosynthesis were also significantly altered due to ethanol exposure (Figure 6B, Appendix A). For example, Tm7sf2, Sqle, Nsdhl, and Dhcr7 are proteins related to steroid biosynthesis and were upregulated (Appendix A) under ethanol condition and supports the increased accumulation of cholesterol derivatives.

### 4.6. Only Metabolomics Identified Glucuronides as the Top-Altered Metabolites by EtOH

A similar analysis of only the combined NMR and LC-MS metabolomics dataset excluded the fatty acyls and phospholipids from the top altered molecular subclasses. This is because the lipidomics dataset highly weighted the contribution from lipids like fatty acyls and phospholipids while the metabolomics dataset emphasized the contribution from other aqueous metabolites. Thus, the pathway analysis of only the metabolomics data set identified glucuronides, amino acids, leukotriene, and monosaccharide as the top-altered metabolites by EtOH (Figure 4A). Notably, leukotrienes are a class of metabolites related to fatty acid oxidation and belong to the eicosanoid family of inflammatory mediators that are produced by the oxidation of arachidonic acid [52]. EtOH administration increased the oxo-dihydroxy-leukotriene B4 but decreased leukotriene E4 (Figure 4D). This observation is likely associated with ethanol induced oxidative stress and the activation of specific cytochrome p-450 enzymes (Cyps) [53]. Indeed, our proteomics results showed that proteins within the arachidonic acids pathway are mostly composed of Cyps that were significantly altered by EtOH (Appendix A).

### 4.7. Ethanol Induced the Glucuronidation of Toxic Metabolites and the Upregulation of Glucuronosyltransferases UGT2s

The metabolomics and lipidomics results also revealed a unique metabolic change induced by ethanol treatment; the selective glucuronidation of a specific set of metabolites (Figure 4A). Glucuronidation is an important process that helps the elimination of potentially toxic xenobiotics (e.g., drugs) and endogenous metabolites. Glucuronidation involves the attachment of glucuronic acid via a glycosidic bond to the substrate. The resulting glucuronide has a higher water solubility than the original molecule and is eventually secreted. Molecular acceptors of glucuronidation are structurally diverse and include lipophilic molecules like steroid hormones, bile acids, biogenic amines, plant and bacterial metabolites, carcinogens, and many therapeutic drugs. We observed an overall increase in the formation of glucuronide-conjugates, such as tyramine-glucuronide and benzoyl glucuronide, after EtOH exposure (Figure 4B). Interestingly, these two metabolites are known to be toxic to hepatocytes and have been previously observed to be glucuronidated [54].

The glycosyl donor in glucuronidation is usually a UDP-hexose, typically UDP-glucuronic acid (UDP-GlcUA), UDP-Glc, UDP-xylose, UDP-Gal, or UDP-GlcNAc [55]. We observed a similar increase in UDP-GlcNAc in EtOH exposed hepatocytes, which may be attributed to a need for more donor molecules to enable the observed increase in glucuronidation. Interestingly, UDP-hydroxymyriostyl-GlcNAc, which requires UDP-GlcNAc, was a highly altered lipid subclass that decreased approximately ten-fold in the ethanol exposed hepatocytes. EtOH administration also decreased the glucuronide-conjugation of sterols (e.g., hydroxy androsterone-glucuronide and dehydrotestosterone-glucuronide). The sterols themselves were also depleted in ethanol exposed hepatocytes. Possibly, as the lipidomics results suggest, the accumulated lipids and sterols were sequestered into the LDs instead of glucuronidation.

These metabolite-conjugated glucuronides are generated by UGTs, which are a highly expressed membrane-bound enzyme family in the liver [56]. The UGTs are classified into four families: UGT1, UGT2, UGT3, and UGT8, where some isoforms, such as UGT1A, UGT2A, and UGT2B are involved in drug metabolism. Conversely, there is no evidence that UGT3 or UGT8 participate in drug metabolism. Again, UGT enzymes facilitate the elimination of xenobiotics and endogenous substrates to protect cells against the toxic by-products of metabolism [57]. Accordingly, UGTs have a broad substrate specificity and can also share substrates [56]. For example, phenols are targeted by Ugt1a9 and Ugt11a6, amines are acted upon by Ugt1a4, aliphatic alcohols are targeted by UGT2a1, bile acids and C18 steroids are acted upon by Ugt2b7 while Ugt2a1 targets C19 steroids. Interestingly, both Ugt1a4 and Ugt2a1 also target C21 steroids. UGTs may also play a role in modulating signaling pathways such as those mediated by steroid hormones [55]. Consistent with these cellular functions, we observed that UGTs were one of the most EtOH impacted protein classes identified from our proteomics analysis of the LDs. Further, glucuronidation (i.e., pentose and glucuronate interconversion) was identified as a significantly affected pathway according to the analysis of our proteomics dataset using either KEGG or WikiPathways (Appendix A). Obtaining the same outcome with two distinct pathway databases further validates the importance of UGTs and glucuronidation as a response to EtOH exposure in hepatocytes. Overall, the UGT superfamily is an important EtOH detoxification pathway [55,58].

While UGT1s were unchanged despite the mRNA levels of *Ugt-1a1* being surprisingly decreased (Appendix A), UGT2s corresponding to Ugt-2b10, Ugt-2b1, and Ugt-2b were significantly upregulated in hepatocytes exposed to EtOH (Figure 6D). These results are consistent with a previous study that observed the UGT2b enzyme subfamily was involved in the metabolic clearance of numerous endogenous and exogeneous compounds, which included bile acids, steroids, drugs, and carcinogens [59]. An enzymatic assay of UGT activity did not show any alteration due to EtOH (Appendix A), but it was likely inconclusive since it measured an overall activity from both UGT1s and UGT2s. In summary, the results from our multiomics study identified UGTs, glucuronides, and the glucuronate interconversion pathway as a major contributor to ethanol exposed hepatoxicity. Notably, UGT2s such as Ugt-2b10, Ugt-2b1, Ugt-2b (Figure 6D) were significantly increased in response to EtOH and warrant further investigation as potential therapeutic targets for ALD.

### 4.8. A Network Map Revealed the Critical Interaction between UGTs and Cyps

A PPI network map of significantly altered proteins and their associated metabolic pathways revealed a major interaction between UGTs and Cyps (Figure 6C). Proteins associated with other top-enriched pathways corresponding to drug metabolism, starch and sucrose metabolism, steroid hormone biosynthesis, and pentose and glucuronate interconversion were also networked to UGTs and Cyps. Like UGTs, Cyps play a vital role in the detoxification of xenobiotics by the liver. Xenobiotic metabolism has two phases. A phase I reaction is intended to increase the polarity and aqueous solubility of the compound while the phase II reaction produces a conjugated compound that can be efficiently detoxified and excreted [60]. Cyps are the key components of the phase I reactions that catalyze the oxidative biotransformation of xenobiotics and other endogenous substrates. As described above, UGTs are an important component of the phase II reaction that produces glucuronide-conjugates [60]. Thus, our observation of an interaction between Cyps and UGTs that are also highly networked with other important metabolic pathways (i.e., drug metabolism, steroid hormone biosynthesis, and glucuronidation) is consistent with the known mechanism for the clearance of accumulated toxic substrates like fats from the hepatocytes.

### 4.9. Ethanol Feeding Directed Carbon Flow into Steroid Biosynthesis and Glucuronide Formation

A single omics technique may not be sufficient to effectively identify the major driving mechanisms behind EtOH-induced fatty liver (LD accumulation) and progressive liver injury. Instead, and as demonstrated herein, the combination of three omics techniques, lipidomics, metabolomics, and proteomics, provided a complementary and system-wide view of the major pathways altered by EtOH and insights into the mechanism of EtOH-induced hepatocellular changes. Metabolomics and lipidomics analysis were combined to create a comprehensive metabolic network (Figure 7), which summarizes the statistically significant effects of EtOH on liver metabolites and lipids identified by NMR and LC-MS. The metabolic network reveals that chronic EtOH feeding increased pyruvate, which is an end product of the glycolysis pathway and also a branch point for fatty acid synthesis, cholesterol synthesis, and the maintenance of the TCA cycle [61]. EtOH feeding also increased TCA cycle intermediates such as succinate and citrate. Notably, this increase in TCA cycle intermediates was correlated with an increase in fatty acid content and its associated metabolites and downstream pathway. Our network analysis also revealed that EtOH feeding increased triglyceride and cholesterol biosynthesis. Interestingly, EtOH-induced cholesterol biosynthesis was associated with a simultaneous downregulation of its glucuronidation. Conversely, EtOH-induced downregulation of bile acid synthesis was associated with an increase in its glucuronidation. EtOH feeding also increased the CDP-choline pathway. Overall, ethanol feeding directed carbon flow through the interconnected pathways of glycolysis, the TCA cycle, and fatty acid metabolism to increase steroid metabolism and alter glucuronidation (i.e., pentose and glucuronate interconversion). This central carbon pathway cascade resulted in the perturbation of specific metabolite-conjugated glucuronides leading to a decrease in sterol-conjugated glucuronides and an increase in other toxic glucuronides such as tyramine-glucuronide and benzoyl glucuronide (Figure 7A).

### 4.10. The Upregulation of UGTs and the Accumulation of Fats and Steroids Induced LD Formation

A summary network map (Figure 7B) was created by combining the metabolomics and lipidomics outcomes (Figure 7A) with the proteomics analysis of the LDs (Figure 6). A multiplatform, multiomics approach was essential to filling in missing data gaps and for uncovering a thorough understanding of a hepatocyte’s response to EtOH as depicted in Figure 7B. In this context, the metabolomics data revealed EtOH-induced alteration to glucuronides and upstream pathways, such as glycolysis, the TCA cycle, and the pentose phosphate pathway. Similarly, the lipidomics data highlighted an overall dysregulation in fatty acids and lipid metabolism that included an upregulation in triglycerides and sterols, which are a major contributing factor to LD formation. The proteomics results further supported the outcomes of the lipidomics and metabolomics analysis where LD membrane protein changes were correlated with the lipid and metabolite changes. Specifically, the proteomics data of the LDs identified an increase in steroid biosynthesis and fatty acid metabolism. Notably, the proteomics results identified the upregulation of UGT2s, the enzymes in the glucuronidation pathway that correlated with the alteration of glucuronides in the metabolomics data set. While a set of LD membrane proteins upregulated after EtOH exposure were found to be associated with the glucuronidation pathway, whether and how this pathway influences the regulation of lipids needs further investigation. Prior reports suggest that fatty acids may also undergo glucuronidation and such modifications may affect the bioavailability of these fatty acids and their derivatives for cellular processes. Interestingly, oxidized fatty acids can be excreted in the form of glucuronides [62,63]. These possibilities and consequences on alcohol induced fatty liver remains to be tested.

## 5. Conclusions

A hallmark feature of ALD is the accumulation of fat within the hepatocytes in the form of LDs. To understand the molecular impact of EtOH on liver parenchymal cells (hepatocytes), we presented the first multiomics investigation that employed both NMR and LC-MS to characterize the metabolome and lipidome changes in hepatocytes and LD membrane proteome changes after EtOH feeding to rats. The detailed system-wide response to EtOH is summarized in Figure 7. Our findings identified fatty acid subclasses as top-altered molecules. Here, we show EtOH feeding specifically increased the levels of diacylglycerols, thia fatty acids, and eicosanoids, to dysregulate lipid homeostasis. EtOH feeding also dysregulated glucuronidation pathway involved in the elimination of drugs, toxins, and endogenous waste. Notably, while EtOH feeding increased bile acid glucuronides, it selectively decreased glucuronides of cholesterols. Interestingly, our findings show that EtOH feeding increased the contents of UDP-glucuronosyltransferase 2 (UGT2) proteins on LD membrane that are known for glucuronidation of sterols. Collectively, our multiomics analyses reveal that lipid metabolism and glucuronidation pathway are dysregulated during the development of fatty liver after alcohol consumption. Furthermore, it identifies specific subclasses of lipids and glucuronides altered by EtOH to provide a basis for further investigation of these metabolites in revealing their roles in the pathogenesis of alcohol associated liver disease.

## Figures and Tables

**Figure 1 biology-12-00028-f001:**
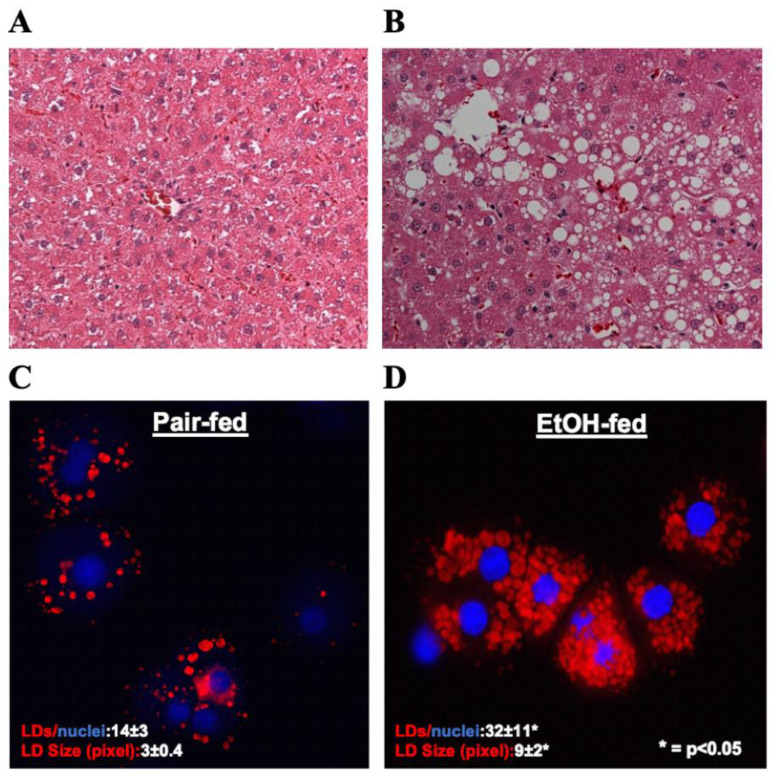
(**A**&**B**) Hematoxylin and Eosin staining was performed on paraffin sections of control and EtOH-fed rat livers. Images are representative of each group. Magnification, 200×. (**C**&**D**) Oil red O staining of LDs in hepatocytes isolated from control and EtOH-fed rat livers. Images are representative of each group.

**Figure 2 biology-12-00028-f002:**
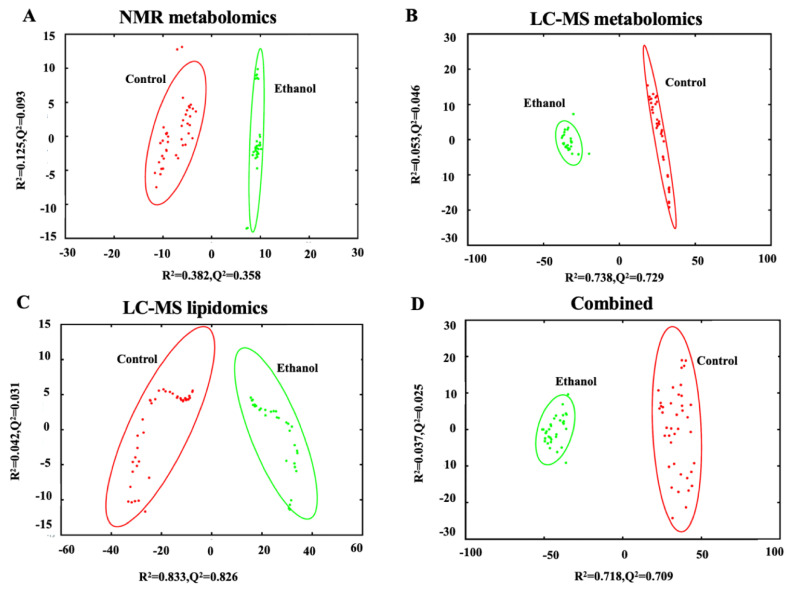
PCA scores plot generated from (**A**) 1D ^1^H NMR metabolomics data set (*R^2^* 0.507, *Q^2^* 0.451), (**B**) LC-MS metabolomics data set (*R^2^* 0.791, *Q^2^* 0.775), (**C**) LC-MS lipidomics data set (*R^2^* 0.875, *Q^2^* 0.857), and (**D**) the combined omics data set (*R^2^* 0.755, *Q^2^* 0.734) comparing control (red) to ethanol (green) treatment of rat hepatocytes. The ellipses represent a 95% confidence limit of the normal distribution of each cluster.

**Figure 3 biology-12-00028-f003:**
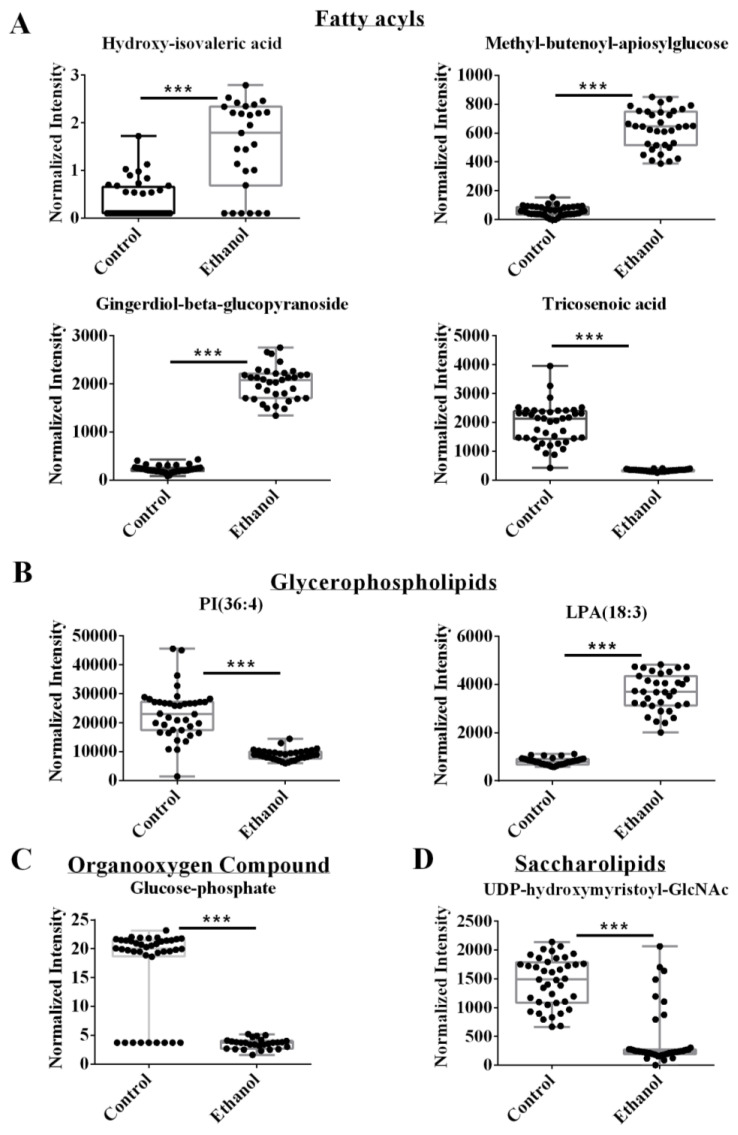
Representative box plots of VIP metabolites or lipids identified from the integrated metabolomics and lipidomics data set with a VIP > 1.19 and an FDR *p*-value < 0.05 using the Benjamini-Hochberg method. Compound class and names are listed above the box plots. *** *p*-value < 0.001.

**Figure 4 biology-12-00028-f004:**
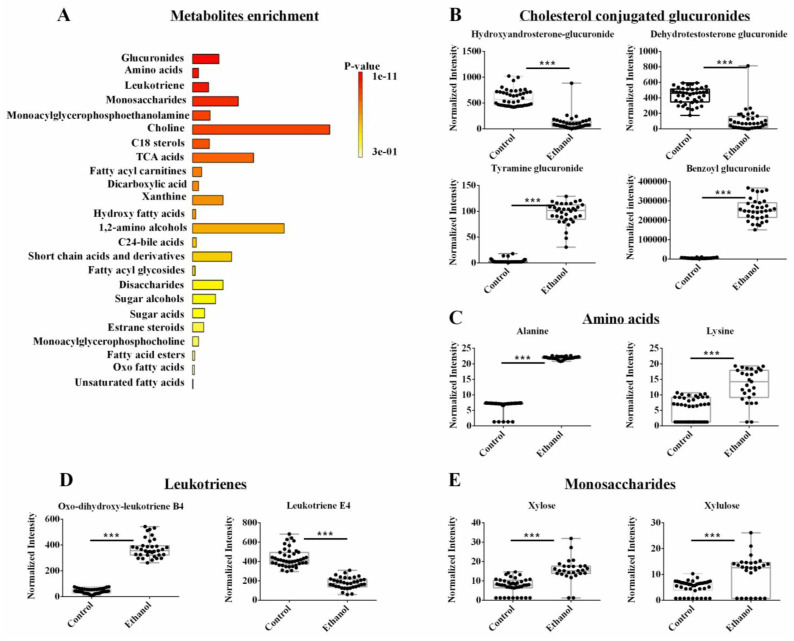
(**A**) An enrichment analysis of the 42 metabolites from the combined NMR and LC-MS metabolomics data set with a statistically significant change in hepatocyte cells following ethanol exposure. (**B**–**E**) Representative box plots of metabolites from the top enriched pathways in (**A**) that highlight the top metabolite changes in the metabolomics data sets. All the metabolites demonstrated a fold change >2.0 and an FDR *p*-value < 0.001 using the Benjamini-Hochberg method. Compound class and names are listed above the box plots. *** *p*-value < 0.001.

**Figure 5 biology-12-00028-f005:**
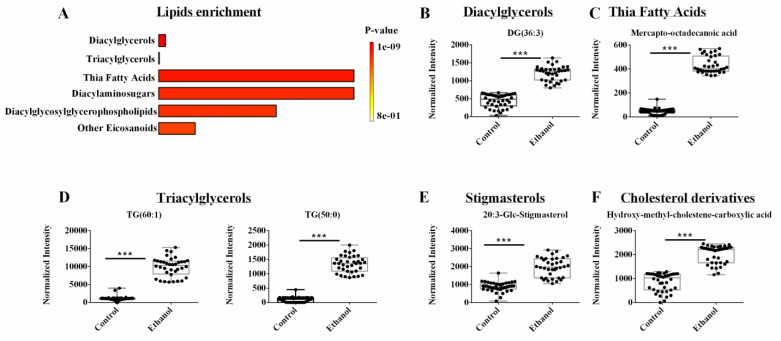
(**A**) An enrichment analysis of the 39 lipids from the LC-MS lipidomics data set with a statistically significant change in hepatocyte cells following ethanol exposure. (**B**–**F**) Representative box plots of lipids from the top enriched pathways in (**A**) that highlight the top lipid changes in the lipidomics data sets. All the metabolites demonstrated a fold change > 2.0 and an FDR *p*-value < 0.001 using the Benjamini-Hochberg method. Compound class and names are listed above the box plots. *** *p*-value < 0.001.

**Figure 6 biology-12-00028-f006:**
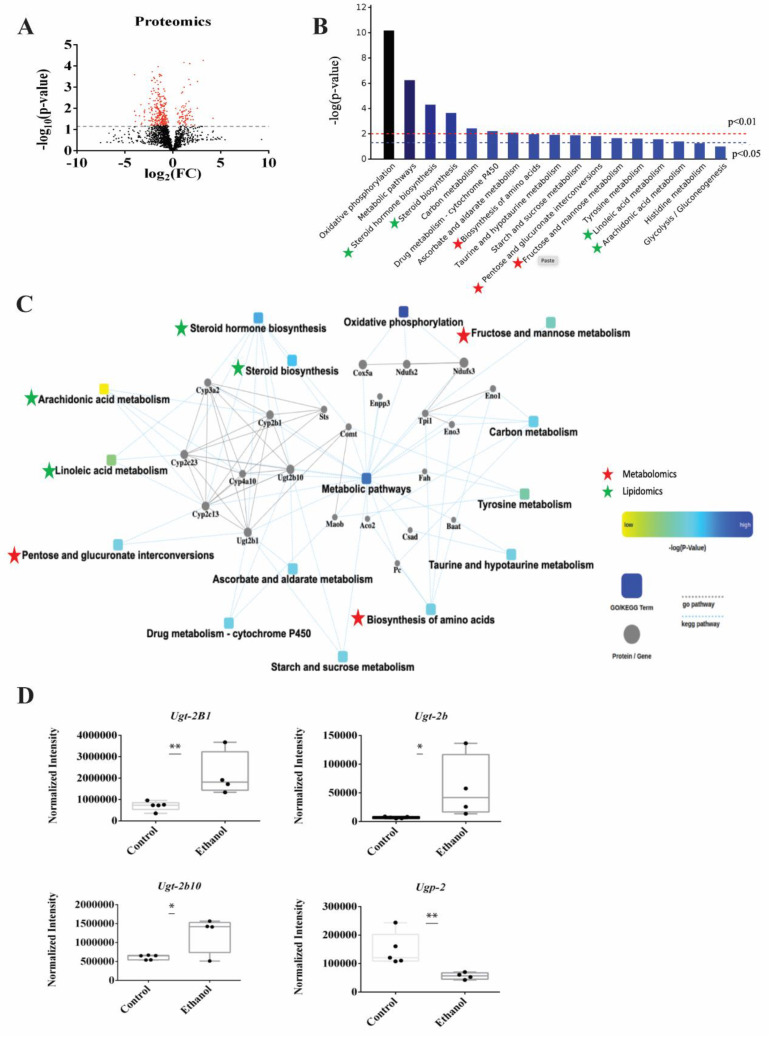
(**A**) A volcano plot summarizing the LC-MS proteomics data set. The dashed line identifies the *p*-value of 0.05. All proteins above the *p*-value threshold are colored red. A total of 2083 proteins were detected with 368 proteins differentially expressed (*p*-value < 0.05). (**B**) A protein enrichment analysis based on the 124 differentially expressed proteins showing the most impactful 17 Kyoto Encyclopedia of Genes and Genomes (KEGG) pathways. The pathways common to both the metabolomics and proteomics data sets or both the lipidomics and proteomics data sets are marked with red and green stars, respectively. (**C**) A Cytoscape generated protein–protein interaction (PPI) network of the significantly changing (*p*-value < 0.05) KEGG pathways. The pathways common to both the metabolomics and proteomics data sets or both the lipidomics and proteomics data sets are marked with red and green stars, respectively. (**D)** Box plots of the UDP-glucuronosyltransferases (UGTs) proteins that are part of the glucuronidation pathway. * *p*-value < 0.05, ** *p*-value < 0.01.

**Figure 7 biology-12-00028-f007:**
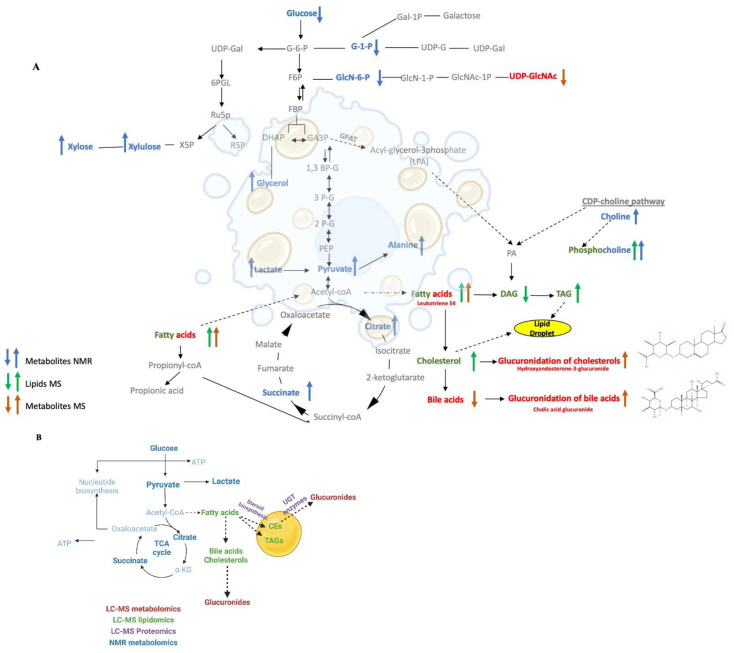
(**A**) A schematic diagram of the metabolic and lipidomic network created from the integrated metabolomics and lipidomics data sets. The network map summarizes the consensus changes in the hepatocyte lipidome (green), NMR detected metabolome (blue), and LC-MS detected metabolome (red) resulting from ethanol exposure. An up arrow indicates a relative increase and a down arrow indicates a relative decrease due to ethanol exposure. (**B**) A summary network map combining the outcomes from the lipidomics, metabolomics and proteomics data set. The network map summarizes the consensus changes in the lipid drop proteins (purple), and the hepatocyte lipids (green), NMR detected metabolome (blue), and LC-MS detected metabolome (red) resulting from ethanol exposure.

**Table 1 biology-12-00028-t001:** Top altered metabolites identified from the integrated metabolomics and lipidomics data set ^a^.

Class	Subclass	Metabolite or Lipid Name	HMDB or Lipid Maps ID	VIP Score	F.C. ^b^	FDR *p*-Value ^c^	MWW *p*-Value ^d^
Fatty acyls	Fatty acids and conjugates	Hydroxyisovaleric acid	HMDB0000754	1.21	4.11	2.10 × 10^−9^	1.58 × 10^−6^
	Unsaturated fatty acids	Tricosenoic acid	LMFA01030091	1.20	0.17	1.74 × 10^−15^	1.71 × 10^−22^
	Fatty acyl glycosides	Methyl-butenoyl-apiosylglucose	HMDB0039952	1.21	10.70	1.62 × 10^−39^	6.32 × 10^−15^
	Fatty acyl glycosides	Gingerdiol -beta-glucopyranoside	HMDB0036123	1.20	8.88	5.84 × 10^−46^	7.74 × 10^−17^
	Fatty acid esters	Dimethylnonanoyl carnitine	HMDB0006202	1.19	0.04	2.15 × 10^−35^	5.71 × 10^−14^
Glycerophos-pholipids	Glycerophosphoinositols	PI (36:4)	HMDB0009899	1.19	0.37	1.90 × 10^−13^	6.15 × 10^−10^
	Glycerophosphates	LysoPA (18:3)	HMDB0114743	1.19	4.60	1.17 × 10^−36^	5.67 × 10^−9^
	Carbohydrates and carbohydrate conjugates	Glucose-1-phosphate	HMDB0001586	1.21	0.22	9.33 × 10^−14^	6.15 × 10^−9^
Pteridines and derivatives	Alloxazines and isoalloxazines	Riboflavin	HMDB0000244	1.21	8.18	9.29 × 10^−50^	2.00 × 10^−22^
Saccharolipids	Diacylaminosugars	UDP-(beta-hydroxymyristoyl)-GlcNAc	LMSL01020003	1.19	0.15	8.85 × 10^−11^	1.70 × 10^−17^

^a^ Selected metabolites had a VIP > 1.19, a fold change > 1.5, and a Benjamini-Hochberg corrected *p*-value < 0.01; ^b^ Fold change (FC) defined as ethanol/control; ^c^ false-discovery rate (FDR) corrected *p*-value using the Benjamini-Hochberg method ^d^ Mann–Whitney U test or Wilcoxon Rank Sum Test *p*-value.

## Data Availability

All data are contained within the article or in the Appendix A. Raw NMR and mass spectrometry data sets are available from the authors and have been deposited in Metabolomics Workbench.

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
