# Peer review of "Multiomics Approach Captures Hepatic Metabolic Network Altered by Chronic Ethanol Administration"

_biology, 2022, doi:10.3390/biology12010028_

Round 1

Reviewer 1 Report

The manuscript submitted by Sakallioglu et al describes a multi-omics approach to investigate disturbances in hepatic metabolism following chronic alcohol consumption. Livers from rats that had been administered ethanol were subjected to proteomic, metabolomic and lipidomic analyses using a combination of high-resolution liquid chromatography-mass spectrometry (LC-MS) and nuclear magnetic (NMR) spectroscopy. The data sets generated were then interrogated with multivariate statistics and network analysis. The authors have reported that exposure to ethanol results in changes in the profiles of neutral lipids and glucuronidation pathways. This an interesting study that offers additional insights into the molecular mechanisms associated with alcoholic liver disease (ALD).

Specific Comments

Methods and Materials: What was the justification of using seven control rats but only six ethanol-fed rats in the study?

Methods and Materials: How does the ethanol diet administrated to the rats compare the typical alcohol consumption in ALD?

Methods and Materials: Provide details of the software package that was used to perform the multivariate statistical modelling.

Results: Outline whether the metabolite and lipid identifications were putative and if they were confirmed against authentic standards.

Results - Figure 5D: How were the fatty acid compositions and positions of the triacylglycerols determined?

Results - Figure 5D: As a plant sterol it would be useful to speculate on the biological relevance of the altered levels of stigmasterol in the rat model.

Reviewer 2 Report

In the current manuscript Sakallioglu  et. Al performed multiomics in etoh fed rat was compared to wild type. Functional analysis of the multiomics dataset showed that rats with ethanol have increase in hepatic fatty acyl content and steatosis. They also showed an increase in glucuronidation tyramine and benzoyl; and a depletion in cholesterol-conjugated glucuronides. Further exposure of ethanol decreases diacylglycerol, and increased triacylglycerol, sterol, and cholesterol biosynthesis. On integration the found that the accumulation of hepatic lipids is due to ethanol modulation of the upstream lipid regulatory pathways, specifically glycolysis and glucuronides pathways. This observation was validated by analysis of LD using proteomics.

The manuscript is well written and is impressive for the details that are provided, though there are some points which needs clarification

1)      Why was A supervised OPLS-DA model was used for the combined metabolomics and lip-idomics dataset were PCA or PLS-DA did not show clear difference

2)      Why didn’t the author prefer simple man Witney test and a volcano plot and instead used opls DA and VIP biomolecules, Ideally simple stats should have provide more clear results or the authors should show no difference in simple stats

3)      Why wasn’t integration i.e correlation clustering of proteomic , lipidomic and metabolomic data performed figure 6C is not integration but enrichment of proteomic , metabolomic and lipidomic pathway of the differentially altered proteins

4)      How from the proteomic data the author directly analyzed only Glucuronidation pathway

5)       Glucuronidation assist in excretion of substances that cannot be used as an energy  why will increase in Glucuronidation result in increase in lipid droplets

6)      Chronic ethanol exposure decreases the Glucuronidation activity but here it is shown increased can author justify increase in Glucuronidation post model development
